# Severe Respiratory Failure Due to Pulmonary BCGosis in a Patient Treated for Superficial Bladder Cancer

**DOI:** 10.3390/diagnostics12040922

**Published:** 2022-04-07

**Authors:** Katarzyna Lewandowska, Anna Lewandowska, Inga Baranska, Magdalena Klatt, Ewa Augustynowicz-Kopec, Witold Tomkowski, Monika Szturmowicz

**Affiliations:** 11st Department of Lung Diseases, National Research Institute of Tuberculosis and Lung Diseases, 01-138 Warsaw, Poland; anna.lewandowska1512@gmail.com (A.L.); w.tomkowski@igichp.edu.pl (W.T.); monika.szturmowicz@gmail.com (M.S.); 2Department of Radiology, National Research Institute of Tuberculosis and Lung Diseases, 01-138 Warsaw, Poland; inga.baranska@interia.pl; 3Department of Microbiology, National Research Institute of Tuberculosis and Lung Diseases, 01-138 Warsaw, Poland; m.klatt@igichp.edu.pl (M.K.); e.kopec@igichp.edu.pl (E.A.-K.)

**Keywords:** bacillus Calmette-Guerin, bladder cancer immunotherapy, BCGosis, BCG pulmonary infection, severe respiratory insufficiency

## Abstract

Intra-vesical instillations with bacillus Calmette-Guerin (BCG) are the established adjuvant therapy for superficial bladder cancer. Although generally safe and well tolerated, they may cause a range of different, local, and systemic complications. We present a patient treated with BCG instillations for three years, who was admitted to our hospital due to fever, hemoptysis, pleuritic chest pain and progressive dyspnea. Chest computed tomography (CT) showed massive bilateral ground glass opacities, partly consolidated, localized in the middle and lower parts of the lungs, bronchial walls thickening, and bilateral hilar lymphadenopathy. PCR tests for SARS-CoV-2 as well as sputum, blood, and urine for general bacteriology—were negative. Initial empiric antibiotic therapy was ineffective and respiratory failure progressed. After a few weeks, a culture of *M. tuberculosis* complex was obtained from the patient’s specimens; the cultured strain was identified as *Mycobacterium bovis* BCG. Anti-tuberculous treatment with rifampin (RMP), isoniazid (INH) and ethambutol (EMB) was implemented together with systemic corticosteroids, resulting in the quick improvement of the patient’s clinical condition. Due to hepatotoxicity and finally reported resistance of the BCG strain to INH, levofloxacin was used instead of INH with good tolerance. Follow-up CT scans showed partial resolution of the pulmonary infiltrates. BCG infection in the lungs must be taken into consideration in every patient treated with intra-vesical BCG instillations and symptoms of protracted infection.

## 1. Introduction

Urinary bladder cancer is the twelfth most frequent cancer worldwide—in 2020 more than 570,000 cases were diagnosed, and more than 200,000 deaths were caused by this disease [1]. The majority of cases are superficial (non-muscular invasive) cancer. Transurethral resection (TUR) of all lesions is the established standard management [2]. Although noninvasive, the disease presents with a high rate of recurrence [2]. Instillations with an attenuated live strain of *Mycobacterium bovis*—bacillus Calmette-Guerin (BCG) are widely used as non-specific immunotherapy significantly reducing the recurrence rate in bladder cancer [3,4,5]. The standard treatment includes six weekly instillations of BCG after TUR, followed by maintenance treatment—three weekly instillations, every six months for three years [5]. The therapy is generally safe and well tolerated, although some side effects, both local and systemic, may be present [6]. The most frequent local complications are cystitis (27–95%), prostatitis (10%) and penile lesions (5.9%) [6]. Systemic complications are much less prevalent and include fever (2.9%), tuberculous spondylitis (3.5%), granulomatous hepatitis (0.7–5.7%), reactive arthritis (0.5–5.7%), and mycotic aneurysms (0.7–4.6%) [6]. Pulmonary *M. bovis* BCG infection and sepsis are extremely rare and occur in 0.4% of patients [6]. 

There are two different hypotheses regarding the mechanisms of disseminated BCG-related disease. It is postulated, that immunological reaction, i.e., granulomatous inflammation without the presence of living microorganisms plays a key role. On the other hand, occasionally positive cultures or positive genetic tests for *M. bovis* BCG are obtained, indicating the presence of active infection. Both mechanisms may coexist. In the biggest report presenting pooled data on BCG treatment complications, published between 1975 and 2013, no specific risk factors for BCG treatment side effects have been identified [7]. The higher mortality was related to older age (≥65 years), disseminated BCG infection and the presence of mycotic aneurysms [7]. The postulated role of immunosuppression has not been clearly confirmed [8] and the guidelines did not recommend different treatments for immunocompromised patients [5].

Identification of *M. bovis* BCG is not straightforward, since it belongs to the *Mycobacterium tuberculosis* complex (MTBC), a highly related mycobacterial species including *M. tuberculosis*, *M. bovis*, *M. africanum*, *M. microti* and *M. canettii* [9]. Live mycobacteria belonging to MTBC, secrete the product of *mpb64* gene—MPT64 protein. Rapid immune chromatography for the detection of the MPT64 protein is a simple and cost-effective method, distinguishing MTBC from non-tuberculous mycobacteria (NTM) [10]. However, some BCG strains have deletions or mutations in the *mpb64* gene. The presence of the *mpb64* gene encoding the MPT64 protein was found in the BCG-Moreau, BCG-Sweden, BCG-Birkhaug and BCG-Russia vaccine strains, whereas it was not found in the BCG-Danish, BCG-Pasteur, BCG-Glaxo, BCG-Tice strains [11]. Therefore, using a test detecting MPT64 protein for identification of the BCG strain with a deletion or mutation in the *mpb64* gene may lead to its misidentification as NTM [11]. 

The treatment of *M. bovis* infection is based on rifampin, isoniazid and ethambutol, using the typical doses as in tuberculosis patients [7]. In some cases, fluoroquinolones or aminoglycosides are used instead of one of the first line antituberculous drugs or as additional drugs [7,12]. The duration of treatment is usually six months if three antituberculous drugs are used and longer if one or two are replaced with fluoroquinolone [7,12,13]. In severe cases with respiratory failure, systemic corticosteroids are used [13]. The treatment outcome is usually good [7]. 

We present diagnostic and therapeutic problems concerning a 75-year-old male with severe respiratory failure in the course of pulmonary BCG infection caused by the *M. bovis* BCG strain that was MPT64-negative.

## 2. Case Report

A 75-year-old male, active smoker (40 pack-years), was admitted to the department of lung diseases in October 2021 due to three-week-long history of hemoptysis, left-sided pleuritic chest pain, general weakness, and fever. The patient was treated with BCG instillations due to superficial bladder cancer for three years, the last instillation was given a month before presentation. In March 2021 he underwent asymptomatic infection with SARS-CoV-2 (positive PCR test performed before an elective hospitalization). He had also a history of esophageal varices, chronic pancreatitis related to cholelithiasis treated with cholecystectomy and partial pancreatectomy, and partial post-traumatic splenectomy. He reported contact with tuberculosis from his father during his childhood. The patient worked as an academic and had no exposure to toxic materials or organic dusts. Before he was admitted to our hospital, he received a course of ciprofloxacin. Ambulatory computed tomography pulmonary angiography (CTPA) showed pulmonary arteries without emboli, emphysema of the lungs’ and discreet nodular, reticular, and ground glass opacities in the peripheral parts of the lungs, which were suggestive of nonspecific interstitial pneumonia (NSIP). 

On admission, the patient presented with dyspnea (respiratory rate 20/min), cachexia (body mass index-19) and generalized weakness. Body temperature was 37.5 °C. Percutaneous oxygen saturation (SpO_2_) on room air was 90%. On auscultation, diminished respiratory sounds were present over the whole lungs, with some bilateral crackles at the basal parts of the lungs. The liver was slightly enlarged. 

Chest X-ray showed pulmonary emphysema, bilateral apical scaring, and some reticular and peribronchial lesions in the lower part of the left lung (Figure 1).

The BD SARS-CoV-2 BD MAX™ real-time RT-PCR test yielded a negative result. Laboratory blood tests revealed slightly decreased number of platelets (106 × 10^9^/L, N: 130–400 × 10^9^), elevated C-reactive protein (CRP) concentration (58.4 mg/L, N: <5 mg/L), N-terminal brain natriuretic pro-peptide (NT-proBNP) concentration (1558 pg/mL, N < 125 pg/mL) and D-dimer (2553 ng/mL, N: <500 ng/mL) as well as increased liver enzymes activity: aspartate aminotranspeptidase (AST) 80 U/L (N: <40 U/L), alkaline phosphatase (ALP) 347 U/L (N: 40–130 U/L) and gamma-glutamyl transpeptidase (GGTP) 218 U/L (N: <60 U/L). Arterial blood gases showed hypoxemia (PaO_2_ 61.7 mmHg, N: 65–90 mmHg) with hypocapnia (PaCO_2_ 30.5 mmHg, N: 35–45 mmHg) and respiratory alkalosis (pH 7.485, N: 7.35–7.45). An ultrasound scan of the abdomen revealed enlarged and high-density liver with uneven margins, suggesting liver cirrhosis. The provisional diagnosis of lower respiratory tract infection was established, and empirical antibiotic therapy (i.e., ceftriaxone), together with oxygen supplementation of 1 L/min through the nasal tube was started. The patient’s clinical condition did not improve—the hemoptysis, low-grade fever and dyspnea persisted, with the need for increasing oxygen supplementation. The blood, sputum and urine cultures were negative.

Microscopic evaluation of the patient’s sputum revealed no acid-fast bacilli (AFB), genetic testing (Xpert MTB/Ultra, Cepheid) for *Mycobacterium tuberculosis* complex (MTBC) was also negative. Taking into consideration the gradual worsening of the patient’s condition, and persistent fever, fiberoptic bronchoscopy was performed, and the bronchial washings were sampled for microbiological tests, including MTB cultures. No signs of bleeding were visible during the procedure.

Antibiotic treatment was modified—ceftriaxone was withdrawn and meropenem with levofloxacin were started. A few days later, dyspnea increased suddenly. The cardiac arrhythmia was noted on physical examination, and the electrocardiogram (ECG) revealed atrial fibrillation. The sinus rhythm recovered spontaneously after a few hours. Despite that, the respiratory failure progressed, and oxygen delivery was gradually increased to maintain SpO_2_ above 90% (maximal oxygen flow was 15 L/min. through the face mask with an oxygen reservoir). A CTPA was repeated and excluded pulmonary emboli. Aortic atherosclerosis without aneurysm and bilateral hilar adenopathy was found. Massive bilateral ground glass opacities in the middle and lower parts of the lungs accompanied by parenchymal infiltrations and bronchial walls thickening were demonstrated (Figure 2)—the lesions progressed compared to the previous CTPA. This presentation suggested alveolar hemorrhage or infection.

Meropenem was replaced with linezolid. Methylprednisolone 60 mg/day intravenously, was started as a rescue medication in a patient with severe respiratory failure, without an identified infectious agent. The patient’s condition gradually improved. At the same time, the positive results of sputum and bronchial fluid cultures on liquid media become available—the acid-fast bacilli (AFB) were cultured.

Isolates of mycobacterial cultures on Mycobacteria Grow Indicator Tube (MGIT, Becton, Dickinson and Co., Sparks, NV, USA) liquid media were not producing MPT64 protein in the immunochromatographic test (Becton, Dickinson and Co., Sparks, NV, USA), indicating the presence of NTM in the patient’s samples. However, stains of isolated cultures showed the serpentine cord formation, characteristic of MTBC [14] (Figure 3). 

To resolve these conflicting results and identify the species of MGIT culture isolates, with morphologic features characteristic of MTBC, but showing negative MPT64 cards and negative MTBC PCR test results, identification was performed using a molecular test (GenoType MTBC VER 1.X, Hain Lifescience, Nehren, Germany). The test confirmed the presence of *Mycobacterium bovis* BCG strain. (Figure 4).

Anti-tuberculous treatment was started (i.e., INH 300 mg/day, RMP 450 mg/day, and EMB 750 mg/day), systemic steroids continued. Further improvement was observed, oxygen therapy was decreased to 2 L/min through the nasal tube, and rehabilitation started. Due to the increase in AST activity to 155 U/L—INH was replaced with levofloxacin 500 mg/day. Final chemosensitivity tests showed the BCG strain resistance to INH, and levofloxacin remained in the treatment schedule. The liver function tests improved, CRP concentration normalized, and further treatment was uncomplicated. The follow-up high resolution computed tomography (HRCT) scan revealed significant partial resolution of ground glass opacities and parenchymal infiltrates, and decreased lymphadenopathy (Figure 5). The patient was discharged home with a three-drug anti-tuberculous regimen including RMP, EMB and levofloxacin. Prednisone was continued in diminishing doses. In the next few weeks, oxygen therapy was withdrawn, prednisone dose was decreased to 5 mg/day. Anti-tuberculous drugs were well tolerated. The treatment of bladder cancer with BCG instillations was withdrawn.

## 3. Discussion

Immunotherapy with BCG is an effective adjuvant treatment for superficial urinary bladder cancer, recommended by the international associations of urologists [5]. The procedure is generally safe; however, a significant number of mostly local adverse reactions causes a big proportion of dropouts from treatment—based on the study of Sylvester et al. only 25–29% of patients receiving BCG instillations completed the three-year treatment regimen [15]. Another study reported systemic side effects in around 30% of patients, who received BCG treatment, with malaise as the most frequent one (15.5%) [16]. Lung infection occurred in less than 1% of subjects [16]. In the most recent literature review, Liu et al. present much lower percentages of systemic complications of BCG treatment–between 0.4 and 5.7% for different entities, again with the lowest incidence of respiratory tract infection [6]. Therefore, in our institution, the tertiary pulmonary hospital, the patients with BCG treatment side effects present very rarely, only if the infection is located in the lungs. As highlighted in the literature, a high grade of suspicion must be maintained to establish a proper diagnosis, as the symptoms may be nonspecific and delayed in time from the last instillation, sometimes even for years [17], although the reported median interval between the procedure and the onset of symptoms was 8 days [7]. Our patient developed symptoms of acute infection a week after the last instillation, nevertheless at that time his complaints were not attributed by doctors to BCG immunotherapy.

The early recognition of BCG-related side effects is difficult. Unfortunately, there are no known risk factors, apart from those related to the procedure itself, such as injury of the urinary bladder during instillation, or acute urinary tract infection at the time of instillation [6,7,15]. Data on clinical case series did not find a relationship between the frequency of complications and patients’ age, smoking status, or immune system disturbances, such as immunosuppressive treatment, chemotherapy, or splenectomy [6,7]. Our patient had a history of chronic pancreatitis resulting from incomplete pancreatectomy and post-traumatic partial splenectomy, without clear indices of immune system impairment. Additionally, he probably had liver cirrhosis with increased portal pressure and esophageal varices.

The radiologic recognition of BCG-related lung disease is also difficult. Lung involvement may present either as a result of the hematogenic spread of *M. bovis*, mimicking “miliary tuberculosis” [7,18,19,20,21,22] or—more frequently—as interstitial pneumonitis with bilateral ground glass opacities [6,13,23]. In children, BCG infection occurs rarely after vaccination and may present as pneumonic infiltrates and lymphadenopathy [24]. Ground glass opacities are usually assessed as hypersensitivity reactions rather than infection [25,26]. In our patient, radiologic changes in the lungs were mostly interstitial, micronodules were few, ground glass opacities and infiltrates dominated in CT scan, hilar lymph nodes enlargement was also present—the whole presentation at the first lung CT scan was interpreted as NSIP. Even at later stages, after lesion progression, the ground glass opacities predomination caused doubts among radiologists, if the BCG-osis is the most probable differential diagnosis. The time of symptom onset in the middle of the severe COVID-19 pandemic and quickly progressive respiratory failure, suggested the possibility of SARS-CoV-2 lung disease, as well. Nevertheless, repeated PCR tests for SARS-CoV-2 were negative.

The correct diagnosis was finally based on the identification of *M. bovis* BCG in sputum and bronchial fluid cultures.

The BCG vaccine was developed as an attenuated live vaccine, derived from virulent strains of *M. bovis* species. The BCG vaccine *M. bovis* strain cannot be differentiated from other members of the *Mycobacterium tuberculosis* complex (MTBC) solely on the basis of phenotypic tests. An example is the immunogenic protein MPT64, which is widely used as a diagnostic marker to differentiate MTBC from nontuberculous mycobacteria. However, bacillus Calmette-Guerin vaccine strains with deletion of the RD2 region do not secrete MPT64. Such culture isolates may be falsely identified as NTM [27,28].

With the introduction of molecular techniques, it has become also possible to differentiate virulent mycobacteria from non-virulent BCG strains. The development of multiplex polymerase chain reaction techniques has provided rapid, sensitive and specific differentiation of BCG vaccine strains [29]. PCR technology identified genomic regions RD1, RD2 and RD3 that were not found in vaccine strains. The loss of virulence was found to be due to a specific regulatory mutation in the RD1 region. This genomic region is present in all virulent human and bovine strains but was not identified in all BCG vaccines [28].

In Poland, two BCG vaccines are used in the treatment of bladder cancer, BCG-Tice and BCG-Moreau. The use of two different strains for immunotherapy in the same population, one with the *mpb64* gene (BCG-Moreau), and the other without (BCG-Tice), further complicates the problem of correct identification. Furthermore, the use of a BCG strain negative for MPT64 will lead to its misidentification as NTM, in mycobacterial cultures [30].

Our patient was treated with BCG-Tice, therefore, the culture isolates were negative for MPT64 protein. This case highlights the importance of careful use of the MPT64 kit.

Positive cultures for *M. bovis* BCG are rarely reported in patients with pulmonary involvement. In the cohort of Perez-Jacoiste Asin et al. urine cultures were positive in three of 11 (27%) patients with two of them having a miliary pattern on chest X-rays [7]. Itai et al. reported positive sputum culture in a patient with solitary pulmonary nodule related to BCG-osis [31]. Positive blood cultures for BCG were reported by Osorio Aira et al. in a patient with a sudden onset of general symptoms immediately after instillation, who developed miliary tuberculosis [21].

Occasionally AFB are also found in lung tissue [32]. We did not take the lung sample for microscopic evaluation, due to increasing hypoxemia. This procedure may help to establish the diagnosis in the absence of MTB in the bronchial washings if the granulomas are found in lung specimens.

There are no established treatment guidelines for BCG infection. A three-drug regimen including RMP, INH and EMB is widely used [33].

In our patient, the additional problem was liver function impairment, probably due to liver cirrhosis. This influenced the treatment, as the anti-tuberculous drugs are hepatotoxic. After worsening liver function tests were noted, we decided to withdraw INH continuing RMP as the most active drug. BCG culture resistance to INH was found and supported our decision. Some strains of BCG have intrinsic resistance to INH and this drug may not be the best first choice in the empirical treatment of BCG infections [34]. In our patient, INH was replaced by levofloxacin, a fluoroquinolone that is active against mycobacteria. Fluoroquinolones, i.e., levofloxacin or moxifloxacin are used in the therapy of mycobacterial lung diseases, in case of anti-tuberculous drug intolerance, resulting in infection eradication [7,12,17,21,31].

The addition of systemic corticosteroids may be necessary for patients with severe respiratory insufficiency in the course of BCGosis [17,35]. Prednisone added to anti-tuberculous therapy in this patient, resulted in the quick resolution of respiratory failure.

## 4. Conclusions

Pulmonary BCGosis is a rare, but important complication of BCG treatment in patients with superficial bladder cancer. It must be taken into consideration in every patient with a history of such treatment, presenting with fever and pulmonary infiltrates, regardless of the time from the last instillation. To identify MPT64-negative clinical isolates of *M. bovis* BCG, it is necessary to use modern molecular methods of identification.

## Figures and Tables

**Figure 1 diagnostics-12-00922-f001:**
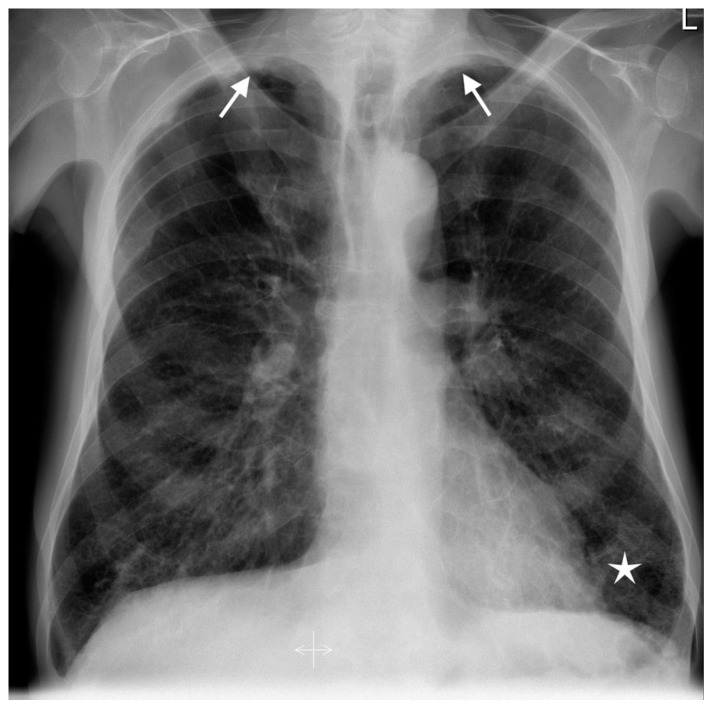
Posteroanterior chest X-ray showing pulmonary emphysema, bilateral apical scaring (arrows), and some reticular and peribronchial lesions in the lower part of the left lung (asterisk).

**Figure 2 diagnostics-12-00922-f002:**
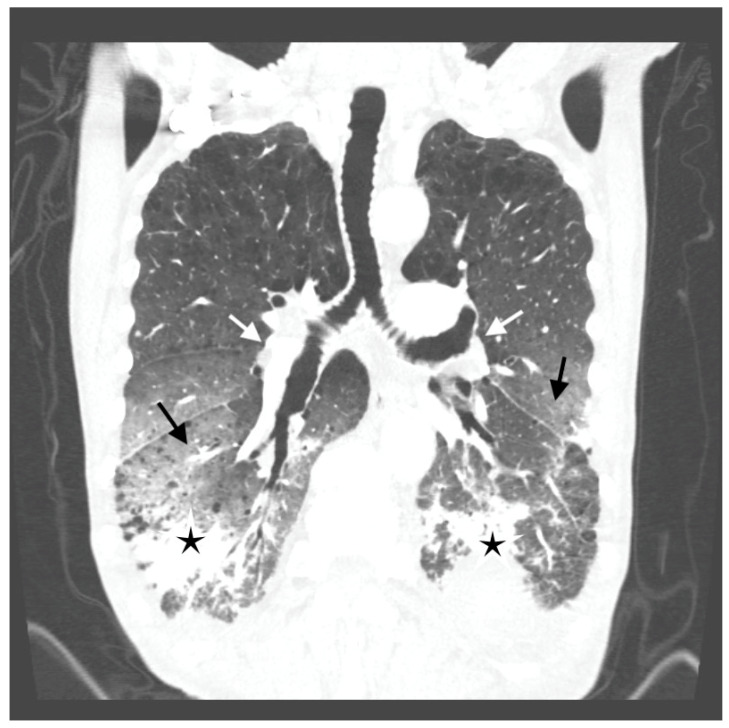
Computed tomography (CT) scan of the chest showing bilateral hilar adenopathy (white arrows), massive bilateral ground glass opacities in the middle and lower parts of the lungs (black arrows) accompanied by parenchymal infiltrations (black asterisks) and bronchial walls thickening.

**Figure 3 diagnostics-12-00922-f003:**
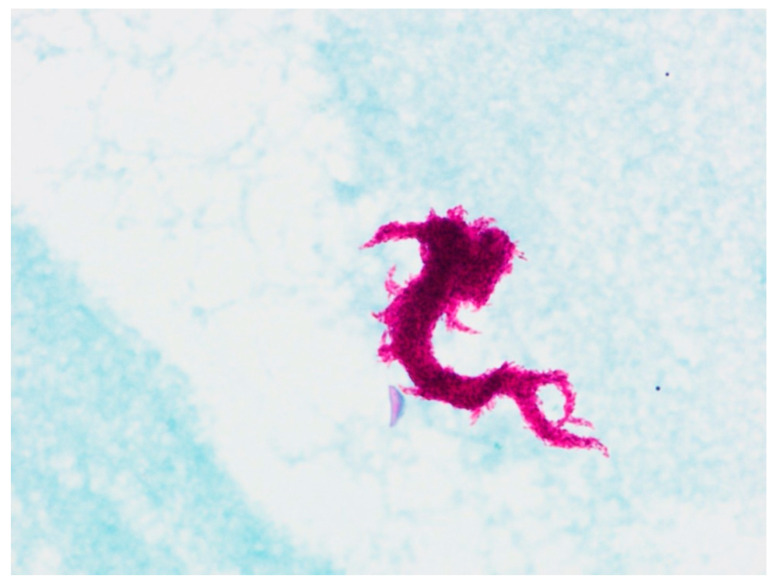
Ziehl-Nielsen-stained slide of mycobacterial cultures obtained on MGIT liquid media with characteristic serpentine cord factor (trehalose 6,6′-dimicolate).

**Figure 4 diagnostics-12-00922-f004:**
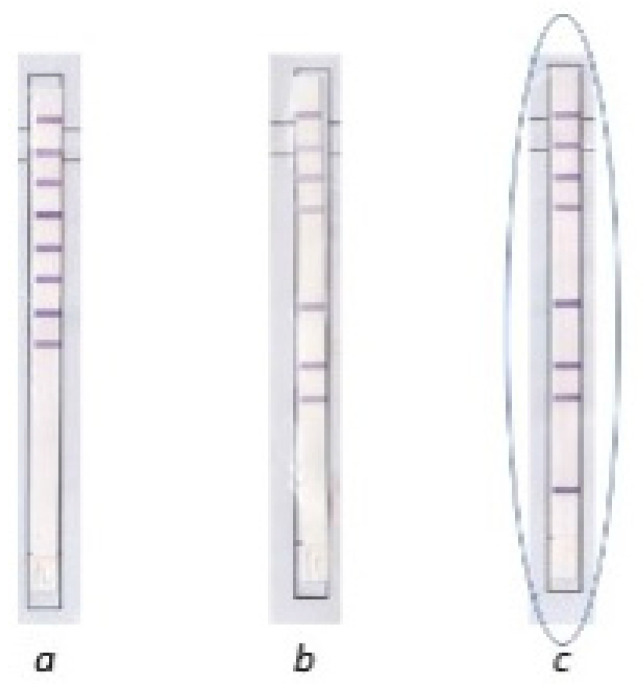
Result of molecular identification of MTBC strains by Hain Lifescience, Nehren, Germany. (**a**) *Mycobacterium tuberculosis*; (**b**) *Mycobacterium bovis;* (**c**) *Mycobacterium bovis* BCG.

**Figure 5 diagnostics-12-00922-f005:**
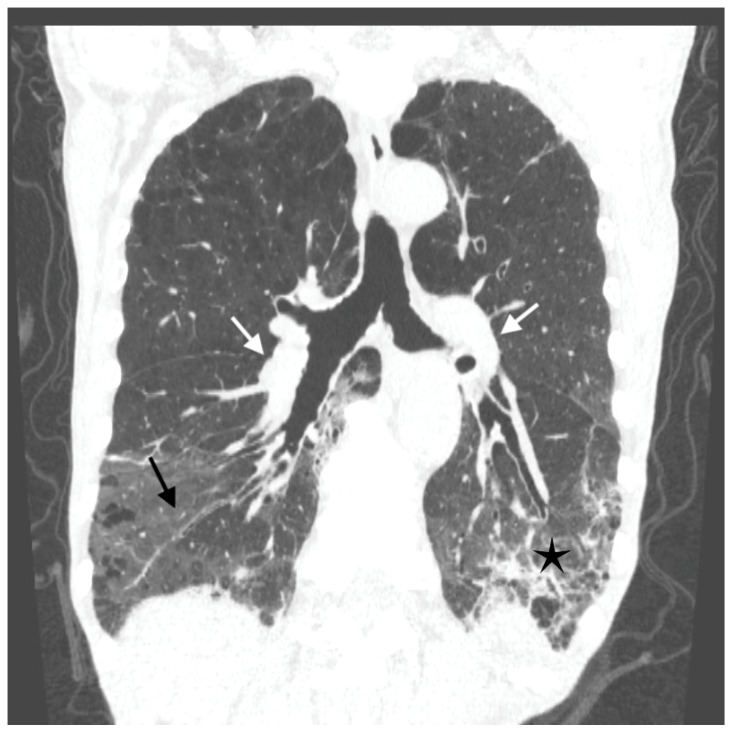
CT-scan of the chest after 3 weeks of anti-tuberculous treatment showing partial resolution of ground glass opacities (black arrow) and parenchymal infiltrates (asterisk), and decreased lymphadenopathy (white arrows).

## Data Availability

Not applicable.

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
