# Peer review of "Severe Respiratory Failure Due to Pulmonary BCGosis in a Patient Treated for Superficial Bladder Cancer"

_diagnostics, 2022, doi:10.3390/diagnostics12040922_

Round 1

Reviewer 1 Report

It is a very interesting case, well-written and organized.  

Minor issues were raised.

  1. How about the role of steroids (dose, duration) in the treatment of systemic side effects such as the current case after BCG Intravesical Instillation therapy?
  2. The common regimen of anti-TB agents in this case, included INH + RMP and EMB., please state in more detail the dosage - dose, and duration. In addition, when we replace INH with levofloxacin, how long will be applied to this patient.

Author Response

Response to the Reviewer 1:

Dear Reviewer,

thank you very much for the clear and thorough review of our manuscript, we are pleased to hear only small changes are needed. Please find our responses to your questions:

  1. The treatment of BCGosis with steroids is based on single cases and case series publications. Most authors used combination of anti-tuberculous drugs and steroids, especially in patients with respiratory failure in the course of BCGosis [Ref. 7, 12, 13, and 35 in the manuscript]. Steroids minimize the inflammatory reaction, that is probably the main cause of respiratory insufficiency. According to literature data, prednisone dose of 40-60 mg/day has been used for few weeks, gradually tapered. Few authors used higher steroid doses in pulses, but the treatment results were comparable.
  2. Standard anti-tuberculous treatment for bovis BCG is the same as for M. tuberculosis and comprises of rifampin (RMP) 600 mg/day in patients with body weight above 50 kg and 450 mg/day in patients weighting less than 50 kg; isoniazid  300 mg/day irrespective of body weight; ethambutol  15 mg/kg/d for 6 months.  There is no consensus concerning the treatment duration in case of using other drug regimens, nevertheless, the authors suggest prolonging therapy to 9 months on such occasion, similarly to the protocol used in the treatment of tuberculosis with M. tuberculosis isoniazid resistance.

Additionally, we added some information about the treatment of BCGosis in the Introduction.

Reviewer 2 Report

Dear Editor,

Thank you for the invitation.

After careful reading, there are some points that authors can further improve this manuscript before it can be accepted for publication.

=============================================

Dear authors,

Thank you for the submission.

After careful reading, there are some points (all are in yellow highlighted colors) that authors can further improve this manuscript before it can be accepted for publication.

Author Response

Response to the Reviewer 2:

Dear Reviewer,

thank you very much for a very thorough and informative review of our manuscript.

We corrected all language and spelling errors indicated in your revision.

As far as other notes are concerned – we added the reference to the indicated sentence on page one [Ref. 11]; we added arrows and other indicators to the figures to make the lesions easier to recognize; we added the commercial names of tests used and we clarified the types of molecular tests.

We explained the abbreviations wherever needed.

We hope the revised manuscript will fulfill the requirements to be published.

Sincerely,

Authors

Reviewer 3 Report

This case report is adequately described.

Author Response

Dear Reviewer,

thank you very much for the comprehensive and positive review. We clarified some issues regarding case description. We hope the new version will be acceptable to you.

Sincerely

Katarzyna Lewandowska

Reviewer 4 Report

Although cases on a similar topic have been published before, this topic is still very rare and interesting.

I think it will give readers a lot of interest and help in clinical practice.

Author Response

Dear Reviewer,

thank you very much for the comprehensive and positive review. We added some new information, we hope that the revised manuscript appears satisfactory to you.

Sincerely

Katarzyna Lewandowska